# Development of Stable Packaging and Producer Cell Lines for the Production of AAV Vectors

**DOI:** 10.3390/microorganisms12020384

**Published:** 2024-02-13

**Authors:** Otto-Wilhelm Merten

**Affiliations:** Miltenyi Biotec SAS, 75011 Paris, France; ottom@miltenyi.com

**Keywords:** stable inducible packaging and producer cell lines, HeLa cells, HEK293 cells, CAP cells, Sf9 cells, rAAV, adenovirus, herpes simplex virus, baculovirus

## Abstract

Today, recombinant adeno-associated virus (rAAV) vectors represent the vector systems which are mostly used for in vivo gene therapy for the treatment of rare and less-rare diseases. Although most of the past developments have been performed by using a transfection-based method and more than half of the authorized rAAV-based treatments are based on transfection process, the tendency is towards the use of stable inducible packaging and producer cell lines because their use is much more straightforward and leads in parallel to reduction in the overall manufacturing costs. This article presents the development of HeLa cell-based packaging/producer cell lines up to their use for large-scale rAAV vector production, the more recent development of HEK293-based packaging and producer cell lines, as well as of packaging cell lines based on the use of Sf9 cells. The production features are presented in brief (where available), including vector titer, specific productivity, and full-to-empty particle ratio.

## 1. Introduction

Adeno-associated virus (AAV)-based vectors are efficient vehicles for in vivo gene therapy in humans. An important set of pre-clinical animal studies have shown their quasi-absence of pathogenicity or toxicity, excellent safety profile, and ability to confer long-term gene expression in a wide range of tissues [1]. In the recent years, an increasing number of clinical trials have been performed in view of the treatment of various rare diseases, including retina [2,3,4], liver [5,6] or neuromuscular diseases [7]. Up to 2023, a total of 350 clinical trials using AAV vectors have been performed, which equates to about 9.1% of all clinical gene therapy trials (http://www.abedi.com/wiley/vectors.php accessed on 20 November 2023).

At the moment of writing this review, seven of these clinical trials have reached regulatory approval by the regulatory agencies (the EMA and/or FDA) for the gene therapy treatment of rare diseases:

These are Glybera (treatment of familial lipoprotein lipase deficiency), approved by the EMA in 2012 [8], but removed from the market in 2017; Luxturna (treatment of Leber’s congenital amaurosis type 2 (RPE65)), approved in 2017 (https://www.chop.edu/treatments/gene-therapy-inherited-retinal-dystrophy-luxturna accessed on 20 November 2023, [9]); Zolgensma (treatment of spinal muscular atrophy (SMA1)), approved in 2019 [10]; Hemgenix (treatment of hemophilia B), approved in 2022 (FDA) and in 2023 (EMA) [11]; Roctavian (treatment of hemophilia A), also approved in 2022 [11]; Upstaza (treatment of severe aromatic L-amino acid decarboxylase (AADC) deficiency), approved by the EMA in 2022 ([12], https://ir.ptcbio.com/node/15111/pdf accessed on 20 November 2023) and probably approved by the FDA in December 2023; and Elevidys (treatment of Duchenne muscular dystrophy), approved by the FDA in 2023 (https://investorrelations.sarepta.com/node/22736/pdf accessed on 20 November 2023).

The AAV vector preparations used for these approved therapies were/are either produced by the HEK293-based transient transfection system or the Sf9/baculovirus system (Glybera, Hemgenix, Roctavian). Several reasons can be identified with respect to the choice of these production methods used for the generation of rAAV vector lots:(i)Since the transient transfection-based production system is the easiest and most straightforward to put into place, and in addition is characterized by its flexibility, most of the AAV vector preparations used for the early stages of product development are produced with this manufacturing method [13]. Furthermore, this production system has been scaled-up to a scale of several thousand liters (e.g., [14], https://www.biopharma-reporter.com/Article/2019/08/22/Pfizer-puts-half-a-billion-into-gene-therapy-facility accessed on 20 November 2023) because in most of the cases there are no time and financial resources available for switching from the rather simple transfection-based production protocol to another production method characterized by elevated scalability, reduced manufacturing costs, etc., as, for instance, provided by stable producer cell lines.(ii)Three of the approved products are produced with the baculovirus system, which shows a much better scalability and productivity, in particular after profound optimization, than the classical transfection-based production protocol [15]. The scale-up of the baculovirus-based production system to several thousand liters is feasible and was shown for the production of influenza viral-hemagglutinin (HA) protein antigens for vaccination purposes [16]. This production protocol is much simpler because the cells have to be only infected with 1–3 different baculoviruses (see below), and it is characterized by reduced variability in comparison to the transfection method. However, when it comes to vector potency, then a mammalian production system might be preferable because it was shown that the post-transcriptional modifications as well as vector methylation are not identical in mammalian and insect cell systems—rAAV vectors produced by the latter system show a reduced in vivo potency [17].Nevertheless, the use of stable rAAV producer cell lines, which can be simply induced by a chemical (without addition of helper virus) to rAAV production, would be preferable because no baculovirus—a contaminant—will be co-produced or need to be removed during purification, and its absence has to be proven in the final rAAV preparation [18].

On the other hand, it is not only the quality feature of the vector product, which is of utmost importance, but also the manufacturing costs and thus the treatment costs because, in the case of whole body/systemic rAAV gene therapy, treatments huge amounts of rAAV vectors are required. These have to be produced at reasonable costs. As an example, in the case of Elevidys (whole body treatment of Duchenne muscular dystrophy), the package insert indicates a dose of 1.33 × 10^14^ vg (vector genomes)/kg of body weight (https://www.fda.gov/media/169679/download?attachment accessed on 20 November 2023). The boys to be treated have an age of 4–5 years, signifying that they have an average weight of 16–18 kg. This means that the required vector amount ranges between 2.13 and 2.39 × 10^15^ vg/patient.

Based on an approximate cell-specific productivity of about 10^5^ vg/c, which can be achieved with practically all production systems (baculovirus system [19], herpes simplex-based production system [20], stable HeLa cell-based producer cell technology [21,22], triple-transfection method [23]) today, a 1000 L reactor culture (with 10^6^ c/mL) can produce (after purification with a yield of 20%) about 2 × 10^15^ vg of purified rAAV, which is more or less equivalent to one patient dose (without taking into account the vector amount required for QC release testing).

This signifies that highly intense large-scale production methods have to be used at the lowest possible costs. Thus, the use of stable cell lines would be the preferable option, in particular when inducible cell lines are used (see below) without plasmid transfection or infection with one or several helper viruses for inducing production of elevated amounts of rAAV vectors.

Further advantages of the use of stable cell lines are that they are fully characterized, allow higher reproducibility, can streamline manufacturing, and, thus, facilitate approval of the rAAV product by regulatory agencies. This, in turn, would lower manufacturing costs, which can result in more affordable rAAV gene therapy treatments (already above mentioned).

This review presents the efforts and developments which had been undertaken to develop stable rAAV packaging and producer cell lines in comparison to other established production methods.

## 2. Different AAV Production Methods (A Short Overview)

### 2.1. Wild-Type AAV

AAVs are small non-enveloped single-stranded (ss) DNA viruses with a diameter of 18–25 nm. They belong to the family Parvoviridae and are classified in the genus Dependovirus. This name is derived from the fact that for their complete replication the presence of a replicating helper virus is required. These can be adenoviruses or herpes viruses. Twelve strains have been isolated from humans and primates and have been characterized, and new serotypes and chimeras, respectively, are continuously discovered or developed [24]. All serotypes share a similar capsid structure, and their genomes have a similar size and organization, i.e., location of the open reading frames (ORF), promoters, introns, poly-adenylation site, and inverted terminal repeats (ITR) (Figure 1). The AAV genome contains two ORFs which encode the four regulatory proteins, the rep proteins (left part of the genome), and the three structural proteins, the Cap proteins (right part of the genome) [25,26]. A further ORF (an alternative ORF of the *cap* gene), encoding for the 23 kDa assembly activating protein (AAP), was described in 2010 [27]. The functions of these proteins are as follows: The two major rep proteins, Rep78 and Rep68, are involved in viral genome excision from the host chromosome, rescue, replication and integration [28], and also regulate gene expression from AAV and heterologous promoters [29,30,31]. The minor rep proteins, Rep52 and Rep40, are involved in accumulation and packaging of replicated ssDNA genomes [32]. The cap ORF is initiated at the p40 promoter and encodes the three structural capsid proteins VP1, VP2, and VP3, which form the capsid with a stoichiometry of 1:1:10 (vp1:vp2:vp3). All transcripts share the same polyadenylation signal. AAP is a critical scaffold that links VPs from their C terminus, promotes viral protein stability, and facilitates VP transport into the nucleus [27,33,34,35]. Finally, the ITRs which flank the AAV genome contain the only necessary regulatory cis-acting sequences required by the virus to complete its infection cycle, namely the origin of replication of the genome, the packaging, and the integration signals. Thus, the ITRs are the only genetic elements which have to be conserved in the vector constructs for transgene replication and packaging [36].

### 2.2. Recombinant AAV Vectors

Transfection-based production methods: Figure 2 presents the basic AAV vector design which is classically used for lab-scale vector productions. The design is quite straightforward. As mentioned above, the ITRs have to be retained and the vector sequence has to be cloned between both ITRs. For vector production, the rep and cap functions as well as the adenoviral helper functions have to be provided. Traditionally, rAAV vectors are produced by transfecting HEK293 cells with three plasmids providing the vector construct, the rep and cap functions, as well as the auxiliary adenoviral genes (*E2A*, *E4orf6*, and *VA RNA* genes). The required adenoviral *E1A* and *E1B* genes are provided by the HEK293 cells [37,38,39]. A double transfection can be used instead. This is based on the use of two different plasmids providing the rAAV vector construct on one plasmid and the rep and cap functions together with the adenoviral helper genes on the second plasmid [40,41].

Despite the advantages of the transfection method with respect to its flexibility—it is easy to switch from one transgene to another one, as well as from one serotype to another one—there are several disadvantages: scale-up of transfection methods ultimately requires laborious and very expensive preparation of large quantities of plasmid DNA [42], resulting in a cost increase; variability in transfection efficiency and a reduced batch-to-batch reproducibility, imposing several challenges to their use in clinical settings [43,44].

Infection-based rAAV production methods: Since transfection-based production methods are characterized by several disadvantages (see above), the use of infection-based production methods represents an option for alleviating most of the transfection-associated disadvantages. Three different methods can be distinguished, based on the use of the (helper) virus used for infecting the production cell line: baculovirus-, herpes simplex-, and adenovirus-based infection.

Baculovirus-based production system: The first baculovirus-based AAV production system was developed by Urabe et al. [45]. It is based on the infection of insect cells (Sf9) which are suspension cells, with three different baculoviruses: one containing the *cap* gene (bac-VP), the other the *rep*78/52 gene (bac-rep), and the last one with the rAAV vector construct (transgene construct flanked by AAV ITRs). Because of several inconveniences related to the genetic instability of the bac-rep construct due to the toxicity of the large rep protein leading to a reduction in rAAV yields over several passages [46], as well as the insufficient expression of the VP1 protein—important for successful transduction of the target cells—several improvements have been proposed [47,48]. Moreover, Smith et al. [48] further improved this production system by reducing the number of required baculoviruses to two, one containing the *rep* and *cap* genes and the second one the rAAV vector construct. Finally, Galibert et al. [49] developed the Monobac system. In this system, all required genes were inserted into one single baculovirus, the *rep-cap* genes into the *egt* locus, and the rAAV vector genome into the classical *Tn7* site.

The reduction in the number of baculoviruses for infecting the Sf9 cells was required because at the used MOI (multiplicity of infection) of 3, as used by Urabe et al. [45], all cells were never co-infected with the different baculoviruses [50]. This led to a reduced production efficiency.

Herpes simplex virus (HSV) 1, another helper virus, was also used for the production of rAAV vectors. Conway et al. [51] developed this system, which is based on two d27.1 replication-defective herpes virus vectors lacking ICP27 expression. Thus, the viral stocks, have to be produced in complementing Vero cells (V27 cells complementing the essential *UL54* gene) [52]. These replication-deficient HSV1 vectors provide the rep-cap genes and the rAAV vector cassette (transgene construct flanked by the ITRs), which are both incorporated into the herpes viral *tk* gene of the UCP27-deficient HSV vector. The rAAV vector production is then performed by co-infecting either adherent HEK293 cells [53,54] or BHK cells grown in suspension [20].

Finally, adenovirus is the helper virus which was used in most of the early-stage rAAV vector productions, when HEK293 cells were transfected with two plasmids (one for providing the rep-cap functions and the second one providing the rAAV vector construct,) followed by infection with adenovirus [55]. As soon as the adenoviral helper plasmid was developed and available, this production system was replaced by the triple-transfection system, simply to remove the co-production of and contamination of the rAAV product by adenoviral helper virus.

However, in the context of the production of rAAV vectors using packaging and producer cell lines, in most cases wild-type or recombinant adenoviruses were used (see below).

## 3. Stable Packaging and Producer Cell Lines

The starting point for the development of stable packaging and producer cell lines was the need to gain access to a scalable rAAV production, which is simpler and more straightforward than the transfection system based on the use of adherently grown HEK293 cells because large-scale production of rAAV vectors for clinical use was not feasible with the existing production protocol at the end of the 1990s.

In the late 1990s/early 2000s, the first scientific papers on the development of stable packaging and producer cell lines were published, which were in most cases based on the use of HeLa cells [21,56], although A549 cells have also been used [57].

Packaging and producer cell lines can be distinguished by the following characteristics. Whereas packaging cell lines contain only the rep/cap functions required for the production of AAV, producer cell lines contain the rep/cap functions as well as the rAAV vector construct (transgene expression cassette flanked by the two ITRs). In more recent developments, inducible producer cell lines have also contained the adenoviral helper functions. Figure 3 presents the development and the mode of AAV production using packaging and producer cell lines (HeLa or A549 cells), respectively.

Despite the fact that due to the expression of the *E1A* gene HEK293 cells represent the best platform for the production of AAV vectors, other cells have had to be used as cellular platform because the constitutive expression of the *E1A* gene, a *rep* transcription activator, leads to the expression of the cytostatic rep proteins. This is the reason why HEK293 cells were initially not considered as a base for the development of a packaging/producer cell line for the production of AAV. HEK293 cells can only be used in conjunction with an induction system for the development of stable packaging/producer cell lines, an approach which was developed more recently (see below).

Earlier in the 2000s, the Sf9 baculovirus system was developed for the production of AAV [10], which was initially based on the infection of Sf9 cells with recombinant baculoviruses. However, in order to also improve this production system and reduce the manufacturing costs, packaging cells have been developed which are going to be used for routine manufacturing of rAAV vectors (see below).

### 3.1. HeLa- and A549-Based Packaging and Producer Cell Lines

#### 3.1.1. HeLa Cell-Based Packaging Cell Lines

In order to induce vector production with packaging cell lines containing the rep/cap functions, the rAAV vector construct can be produced either via transfection or via infection by an adenovirus–AAV hybrid virus (preferable) and infection by the adenoviral helper virus. In general, authors have used a sequential infection scheme of the packaging cell lines, initiating this infection with the adenoviral helper virus with functional genes of the E1 region and subsequently (e.g., 24 h later) with an adenovirus-AAV hybrid virus containing the rAAV transgene sequence in the E1 region of the adenovirus [58]. This sequential infection is required because it allows for the production of sufficient rep proteins due to the activation of the AAV p5, p19, and p40 transcription units of the *rep-cap* genes by the adenoviral helper virus before replication of the E1-deficient rAd genome is initiated.

All studies have established a strong correlation between rep/cap gene amplification and a high yield of rAAV production [59,60,61].

Although a wt adenovirus can be used for the initial infection, it is preferable that this adenovirus is replication deficient in order to avoid its co-production in parallel to AAV and contamination of the final AAV vector product. In this context, adenoviral mutants characterized by a temperature-sensitive *E2b* gene (sub100r [62] or ts149 [60]) can be used. Furthermore, the adenovirus–AAV hybrid virus is preferentially E1 negative in order to preclude its production during the AAV production and, thus, contamination of the final AAV vector preparation.

The advantages of the packaging cell line-based production systems are the fact that these systems are rather versatile and the production of new vector/new transgene constructs can be easily performed via the modification of the adenovirus–AAV hybrid-virus. Further advantages are the 5–10-fold higher vector productivity in comparison to the transfection production method [63], as well as the fact that no rcAAV (replication competent AAV)/wtAAV is generated in parallel to rAAV production (<1/10^9^ AAV vector particles) [62,63]. The generation of rcAAV is a general problem observed in the transient transfection-based production system because of potential non-homologous recombination events during transfection [38,64].

However, some drawbacks should also be mentioned here: in order to avoid the production of adenovirus helper virus-free AAV batches, temperature-sensitive strains have to be used, which unfortunately are prone to reversions and are thus difficult to produce and characterize. Furthermore, in order to produce AAV with this production method, the packaging cells have to be infected with two different viruses, which signifies that, for a GMP process, two different adenoviruses have to be produced under GMP conditions and undergo a QC release testing. For this reason, large-scale productions are preferably performed using producer cell lines because only one adenovirus is required for inducing the production of AAV vectors (mono-infection in contrast to a staggered infection with two adenoviruses).

In the context of the use of packaging cell lines, the recently published TESSA (Tetracycline-Enabled Self-Silencing Adenovirus) should be mentioned. This system that was initially developed for an infection-based rAAV production system is based on the following modifications of the adenoviral helper viruses [65]: the helper adenovirus self-inhibits its major late promoter (MLP) to truncate its own replication. The insertion of a tetracycline repressor (TetR) binding site into the MLP and encoding of the TetR under its transcriptional control allows for normal virus replication in the presence of doxycycline but only genome amplification and the expression of early genes (the helper functions) in its absence. By co-infecting HEK293 cells with two TESSAs, one coding for AAV *rep* and *cap* genes and the second one for the rAAV genome, the two TESSAs delivered adenoviral helper functions in addition to the AAV *rep* and *cap* genes and the rAAV genome. Up to 30-fold-more rAAV vectors were obtained compared to the helper virus-free plasmid approach. Cell-specific production levels of >10^5^ gc/cell have been reported. This production system has significantly improved particle infectivity (5–60 fold) as shown for rAAV2, 6, 8, and 9. Moreover, the reduction of 99.99992% compared to normal adenovirus production is a further advantage because of the very low contamination of the rAAV preparation by the helper virus.

In 2023, the same authors [66] published an adaptation of the TESSA system to use in the context of packaging cell lines containing the *rep* and *cap* genes (HeLa RC32). The original TESSA system enables rAAV production only when the *rep* and *cap* genes are provided *in trans*, but not from stable packaging cells. Using HeLa RC32, the authors have shown that expression of the adenovirus L4 22/33K unit is essential for rep/cap amplification but that the proteins are titrated away by binding to replicating adenovirus genomes. siRNA-knockdown of the adenovirus DNA polymerase or the use of a thermosensitive TESSA mutant (TESSA-E1-tsDNA) decreased adenovirus genome replication whilst maintaining MLP repression. In this study, the HeLa RC32 cells were either transfected with the pAAV transgene plasmid or infected with rAAV transgene in parallel to infection with modified TESSA. Thus, this paper presented a development of the TESSA system towards the use for rAAV production using packaging cells; however, further modifications are still required to render this system of interest for the industrial production of rAAV. Amongst other issues, it will be necessary to deliver the rAAV genome directly via TESSA and not separately via transfection or rAAV infection. However, in fine, this system is of high interest because it is an infection-based system (which is always more efficient than the transfection-based production system [59]) which is characterized by a or absence of co-production of contaminating helper virus.

The development of a packaging cell line was in many cases also the intermediate step towards the establishment of a producer cell line.

#### 3.1.2. HeLa Cells versus A549 Cells

In 1998, Gao et al. [62] published the development of HeLa-based packaging cells. Clones were generated by transfecting HeLa with plasmids containing the rep/cap helper functions under the control of AAV promoters. A number of 708 clones were selected, of which 515 survived expansion. However, only eight of these clones were able to trans-complement rep/cap. The best clone, called B50, was further evaluated. In order to induce the production of AAV vectors, the cells were infected with an adenovirus defective in *E2B* to induce the expression of the rep and cap proteins, which was followed by infection (24 h later) with a replication-defective hybrid adenovirus in which the AAV vector construct was cloned into the E1 region. This infection led to a 100-fold amplification and rescue of the AAV genome, leading to a high yield of recombinant AAV, free of replication-competent AAV; 3.3 × 10^8^ transducing units and 6.4 × 10^12^ genome copies were obtained per 10^9^ cells.

Since HeLa cells contain sequences of the human papilloma virus, the same group evaluated the A549 cells using the same approach [67]. A549 cells were derived initially from a human alveolar cell carcinoma without any viral-mediated transformation [68]. A549 cells were transfected with the P5 *rep*/*cap* and clones were selected. One clone (K209) led to a 1000-fold amplification of the *rep*/*cap* genes and a high-level expression of AAV vectors upon infection with adenovirus. The required adenoviral MOI was 5–10-times lower than that required for the induction of AAV production by the HeLa-based packaging clone [62,67]. The vector yield per cell was comparable to that obtained with B50 cells, meaning that both cell lines can be used for the production of AAV; however, there is an advantage for the A549 cells because a lower MOI of adenovirus is necessary for infection. Furthermore, these results also indicate that the presence of papilloma viral factors present in HeLa cells are not required for AAV production.

It should also be mentioned here that only those clones that reached a level of >3000 copies of *rep* genes per cell produced high levels of AAV vectors [67].

#### 3.1.3. Producer Cell Lines

AAV producer cell lines contain the rep/cap helper functions as well as the rAAV vector construct. Their development can be sequential, starting from a packaging cell line into which the rAAV vector construct is either introduced via stable transfection [56,59,69,70] or via transduction with the rAAV vector [57,71]. In the case of plasmid transfection, the development of such a cell line can also be performed in the form of a ‘one shot’, meaning that all required functions are inserted into the producer cell line at the same time [22,56]. The advantage of plasmid transfection relates to the possibility to select transfected cells via a resistance marker cloned into the plasmid. In the case of transduction using an rAAV vector (up to three rounds of transduction at 10^5^ vp/c were used), the selection process based on PCR screening and rAAV vector-production capacity induced by adenovirus infection can be rather cumbersome [57]. Martin et al. [22] reported that less than 6% of all cells analyzed are high-rAAV producer cells, meaning that a majority of the cell clones are non-producers. This also signifies that a large number of clones have to be analyzed.

For inducing AAV production in producer cell lines, the producer cells are infected at optimal cell density with the adenoviral helper virus, which is either replication competent or replication incompetent (see also packaging cells). The use of a replication-competent adenovirus for inducing rAAV production is straightforward and easier to produce than a replication-incompetent adenovirus; however, it has to be kept in mind that adenovirus is co-produced with the rAAV vector and has to be removed during DSP. Furthermore, for clinical vector productions, the removal and inactivation of the helper adenovirus has to be validated via a clearance study [21,72] (see below). In order to alleviate this drawback, Farson et al. [57] have used a replication-incompetent adenovirus; however, these viruses are sometimes characterized by instabilities [69] and, often, the obtained rAAV titers are lower than when replication-competent adenoviruses are used [73].

The used adenoviral MOI ranges from 10 to 100 and leads to considerable amplification of the rep and cap genes (by a factor of 100) [59]. Higher MOIs (optimum: 100) are required in order to ensure a sufficient rep-cap amplification, which directly impacts AAV vector yields [59].

Summarizing, the critical characteristics of stable rAAV producer cell lines are the following: (i) the integrated rep-cap and rAAV vector sequences have to remain stable over the passages without drug selection; (ii) the rAAV promoters have to be silent during sub-cultivation and cell amplification; and (iii) upon infection with helper virus (adenovirus or HSV1), a high-level expression and amplification of rep and cap genes has to be induced, and the rAAV genome has to be rescued and replicated.

##### Use of Different Helper Viruses: Adenovirus Versus Herpes Simplex Virus 1

Using stable HeLa-based producer cell lines, Toublanc et al. [70] have shown that the use of HSV1 at an MOI of 5 was equally efficient in inducing rAAV vector production as a wild-type adenovirus used at an MOI of 50. Furthermore, the production kinetics were different for both helper viruses, with the maximum rAAV yield obtained after 24 h in the case of HSV1, whereas 48 h were required for the adenoviral infection. Moreover, these authors have shown that HSV1 mutated for the UL9 or the UL30 genes were as efficient as wild-type HSV1 for inducing rAAV production (at a small scale). The comparison of rAAV produced via HSV1 infection and via HSVΔUL30 in vivo showed equivalent potency/efficiency.

The advantages of the use HSV1 or a mutant of HSV1 for inducing rAAV production by producer clones are the kinetics of vector production as well as the fact that the required MOI is much lower than that used for the adenovirus infection, which in principle would speed-up the production and reduce production costs. The disadvantages on the other hand are the much-higher pathogenicity of HSV1 as well as the fact that mutant HSVs are more difficult to produce.

##### Use of the Producer Cell Approach for the Production of Large rAAV Vector Preparations

Optimization and scale-up studies for the large-scale production of AAV vectors in suspension were performed using stable HeLa-based producer cell lines.

A review paper by Thorne et al. [21] described the development of an rAAV vector-production process based on stable producer cell lines. The cells which contained the rep/cap genes were transfected with a plasmid containing the rAAV sequence, which was followed by selection and screening in order to generate the final cell clone, which was then established in a GMP bank. The final selection criteria were >50,000 DNase-resistant particles per cells, >50% of full capsids, rcAAV below detection level, and the produced vector had to have the correct identity and appropriate infectivity. The stability of the cell bank was then validated for a minimum of 60 populations. The serum-free suspension-based production process was scaled up to 250 L [21] and further to a 2000 L scale (Thorne et al. [14], and Celladon Corporation conducted an initial scale-up of manufacturing process for MYDICAR up to commercial scale, 5 January 2015). Another paper by Thorne et al. [72] described the methods which had been used for the inactivation and removal of contaminating adenovirus. The combinations of heat treatment (52 ± 1 °C) and nanofiltration led to adenovirus clearances of 5–6 and >6 log10, respectively.

Martin et al. [22] have also published on the development of stable producer cell lines for clinical vector production. In contrast to Thorne et al. [21], they used a single plasmid containing three components: the vector sequence, the AAV rep and cap genes, and a selectable marker gene, which was stably transfected into HeLaS3 cells. This approach was also initially used by Clark et al. [56] for the establishment of the first stable rAAV producer cell line. Martin et al. [22] reported productivities ranging between 5 × 10^4^ and 2 × 10^5^ DNase-resistant particles/cells stable over >60 population doublings. Furthermore, the cells contained between 12 and 50 plasmid copies. Upon adenovirus infection, the rep/cap copy number was amplified about 100-fold. The vector preparations contained >70% of full particles and the preparations were negative for rcAAV (below 0.0002% = below limit of detection). The authors only presented the results obtained from max. 1 L production cultures, but these cells were used for large-scale rAAV vector productions.

### 3.2. HEK293- and CAP-Based Packaging and Producer Cell Lines

The use of HEK293 cells as base for the development of rAAV packaging and producer cell lines is less straightforward because the constitutive expression of the *E1A* gene, a *REP* transcription activator, leads to the expression of the cytostatic rep proteins (via activation of the p5 promoter). This signifies that HEK293 cells can only be used in conjunction with an induction system for the development of stable packaging/producer cell lines (as already indicated).

The first HEK293-based rAAV producer cell line was developed by Qiao et al. [74]. The authors developed a helper virus-free inducible producer cell line containing the sequences of the rAAV vector construct, the cap genes, as well as the four rep gene sequences, which were conditionally disrupted by insertion of an intron that harbors transcription-termination sequences flanked by the LoxP sites into their shared coding region. The promoters of the rep gene proteins are not affected; however, the presence of this intron leads to a premature transcription stop of rep transcription (Figure 4A). Upon infection by an adenovirus (MOI = 5) deleted for E1A, E1B, and E3 and carrying the *cre* gene, these producer cells started to generate high titers of AAV vectors (in this specific case with GFP as transgene) due to the reactivation of the transcription of all four rep proteins. This switch system was initially tested with the lacZ gene, and a 600-fold induction of β-galactosidase activity was observed. Finally, the authors evaluated this induction system for rAAV coding for other transgenes. The generated HEK293-based producer cell lines showed normal growth characteristics, high stability, and high yields of rAAV vectors. These producer cell lines did not produce rcAAV—no rcAAV particle could be detected in up to 10^9^ viral genome particles. The cells produced about 50% of full rAAV particles with a significantly higher infectivity when compared to the triple-transfection method and a HeLa cell-based packaging cell line.

The main drawback of this producer cell line was the fact that the development of these cells required multiple cloning steps for the vector and packaging plasmids, and a two-step transfection and selection for stable cell lines. Thus, based on this approach, the same group [75] simplified the development of such producer cells by a one-step cloning of AAV vector cassette into the serotype-specific packaging plasmid and a single-plasmid transfection and selection for stable AAV vector producer cell lines (the cloning procedure was streamlined by using the Gateway technology). Upon infection with an E1A/E1B-deleted helper adenovirus, these producer cells produced high yields of different AAV serotypes (2, 8, 9): 5–8 × 10^13^ vg per Nunc Cell factory (0.9–1.3 × 10^5^ vg/cell). The different clones contained 10–50 *rep/cap* gene copies/cell, and upon infection the copy number was amplified about 10–20-fold.

Although rAAV production was performed with adherently grown HEK293 cells, the switch to serum-free suspension culture should not be too difficult because HEK293 cells can easily be adapted to suspension growth.

In 2023, another inducible HEK293-based rAAV producer cell line was published [76]. These cells contain the inducible *rep* gene constructs, the sequences for the expression of the cap proteins, as well as adenoviral helper genes (E2A, E4, VA-RNA). The expression of rep40 and rep68, inserted into the HEK293SF cells using lentiviral vectors, was under the control of two inducible promoters with different expression levels (higher expression levels for rep40 and lower expression level for rep68). The expression was induced by the addition of cumate/coumermycin.

Three of the established clones produced vector levels comparable to the triple-transfection process; the vector-production levels were very similar for rAAV2 and rAAV DJ. Furthermore, the clones were stable in continuous cultivation for up to 7 weeks. However, it should be mentioned here that during later passages, a reduction in the rep68 levels was observed, leading to a reduction in vector titers. This reduction in the expression was probably caused by an epigenetic silencing mechanism. Thus, this rAAV production system is not yet ready for large-scale use.

Based on previous studies [77], Lu et al. [78] have developed another inducible HEK293-based rAAV producer cell line by optimizing helper constructs and reducing the expression of the transgene during vector production. The different functions were placed on three separate plasmids which were transfected into HEK293 cells to generate stable rAAV producer clones. These plasmids contain the following functions (Figure 4B): the rAAV vector plasmid contains the rAAV vector construct flanked by the two ITRs with the expression of the transgene driven by the RSV-LacO promoter. The vector construct was cloned into a transposon backbone with a *lacI* repressor gene linked to a puromycin-resistance gene driven by a phosphoglycerate kinase promoter, thus leading to an inducible rAAV genome construct. The second transposon construct (the so-called replication module) provides the large rep protein (rep68) tagged with a mutant destabilization domain (FKBP12) as well as the adenoviral (Ad2) helper proteins (E4orf6, DBP), whose expressions are controlled by the Tet-on-based induction system. Furthermore, this replication module contains a hygromycin-resistance gene linked via T2A to the reverse Tet transactivator. The third transposon construct (the so-called packaging module) provides the sequences for expressing the capsid proteins as well as the small rep protein (rep52), whose expressions are controlled by a cumate switch. Although these cells already produce better titers than the initially developed cells [77], only the addition of the proteasome inhibitor MG132 for reducing the host cell-initiated degradation of rAAV allowed for the production of rAAV levels comparable to the triple-transfection system. Furthermore, 10^5^ capsids and than >10^4^ vg are produced per cell. With respect to the level of full particles, they ranged between 25% and >50%.

**Figure 4 microorganisms-12-00384-f004:**
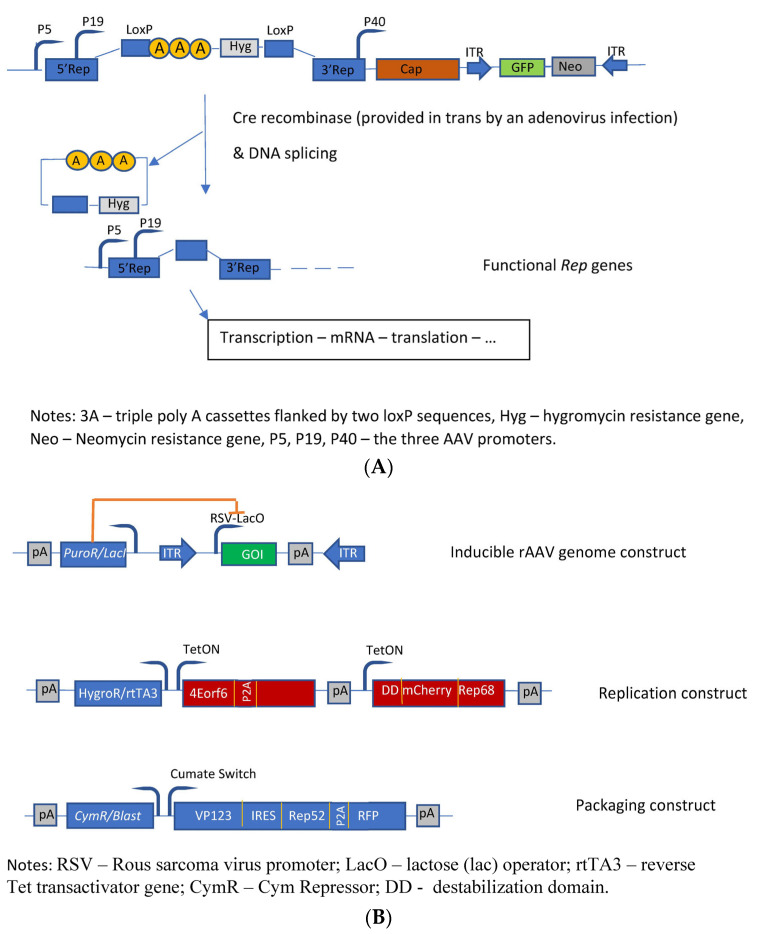
Molecular constructs used for the development of the different HEK293 cell based packaging/producer cell lines. (**A**) Developments performed by Qiao et al. [74]: Construction of dual-splicing switch AAV packaging plasmid; the switch system is based on the Cre-Lox system; (**B**) Constructs used for the development of an AAV producer cell line [78].

In view of improved vector production, the authors proposed a further reduction in the expression of the transgene during vector production (e.g., use of tissue specific promoters whose choice depends on the target tissue to be transduced) as well as an increase in the expression of capsid proteins combined with an improved efficiency of viral genome packaging into capsids. A further improvement in production might be achieved via an exploration of other adenoviral helper components or their combination.

Finally, CEVEC (now part of Cytiva) has developed inducible rAAV producer cell lines based on the use of HEK293 and CAP [79,80] cells. The so-called Alpha cell line contains all the required (helper) functions (adenoviral helper functions: *E2A*, *E4orf6*, *VA RNA*; the AAV *rep* functions), which are inducible upon the addition of doxycycline (Tet-on induction system). In the second step, the desired serotype-specific *cap* sequence as well as rAAV vector sequence are stably integrated in order to obtain to the specific rAAV producer cell line. Genome integration of the *rep* gene was possible because of the inactivation of the p19 promoter while the functionality of Rep78/68 was maintained, patented by CEVEC [81]. A clonal stability of over 75 passages was communicated. Unfortunately, no manuscript has been published describing the details of this cell line development. This production system was developed in view of large-scale industrial production of rAAV for clinical and commercial purposes (→ELEVECTA AAV production platform). RAAV production processes have been scaled to 200 L. When inducing rAAV production at a cell density of 5–6 × 10^6^ c/mL, the cell culture produces vector titers of up to 5.7 × 10^13^ vg/L (6 days after induction) [82]. Further optimization of the cell culture process towards a high cell density perfusion system allowed for obtention of 2 × 10^7^ c/mL and an rAAV production level of >10^15^ vg/L with about 30% of full particles [83].

### 3.3. Insect Cell-Based Packaging Cell Lines

As mentioned above, another approach for producing large amounts of rAAV vectors is the Sf9/baculovirus system. The original baculovirus system was based on the infection of Sf9 cells with three baculoviruses [45], followed by optimization in order to improve the expression system but also for reducing the number of baculoviruses via the development of the dual baculovirus system [48] and the Monobac system [49].

As mentioned, the reduction in the number of baculoviruses was necessary, firstly for reducing the number of different viruses to generate for production purposes, but also for ensuring that all Sf9 cells received the required functions for producing rAAV.

The development of the Monobac system [49] was already mentioned; another approach is the development and optimization of stable producer cell lines, which is presented in the following.

The initial development was published by Aslanidi et al. [50]. The authors developed a two-component system consisting of one recombinant Sf9 cell line and a recombinant baculovirus. The recombinant Sf9 cell line contained silent copies of the rAAV *rep* and *cap* genes (Figure 5A). Both genes were controlled by the very late baculoviral polyhedrin promoter and the cis-acting enhancer element hr2-0.9. Upon infection with baculovirus harboring the rAAV genome, the hr2.09 enhancer was transactivated by the baculoviral *IE-1* gene, leading to the expression of the *rep* and *cap* genes of AAV. A feed-forward loop was initiated by the induced Rep78 protein, boosting the amplification of the integrated genes by interaction with the incorporated cognate AAV rep-binding site (RBE) because the P5 RBE mediated the amplification of integrated AAV sequences in mammalian cells [84], thus dramatically improving yields of rAAV production, as shown for HeLa-based packaging cell lines [84]. Furthermore, it was also shown previously that either AAV ITR or P5 RBE modulates the expression from the P19 promoter [85]. In the system developed by Aslanidi et al. [50], the result was an elevated production of rAAV which could exceed the production levels obtained by the triple-transfection method by-10 fold. These packaging cell lines have been evaluated for the production of AAV serotypes 1 and 2. A further advantage of this system is the genetic stability of these cells.

Mietzsch et al. [86] further developed this system to the OneBac 1.0 platform for the generation of all AAV serotypes ranging from 1 to 12 (Figure 5B). Specific production rates of up to 5 × 10^5^ infectious particles per cell (AAV3) were obtained.

Although these packaging cells are of high interest because of the simplification of the insect cell/baculovirus-based production system, unfortunately they are affected by an important disadvantage. The collateral packaging of helper DNA into AAV capsids was observed. This disadvantage was tackled by further developments performed by Mietzsch et al. [87].

Mietzsch et al. [87] reported considerable co-packaging of *rep*, *cap*, *bsd,* and *hr2* sequences into rAAV5 capsids, e.g., reaching up to 35.6–55.3% of particles positive for the *cap* sequence. They have shown that the removal of the *RBE* sequence from *cap* and *rep* constructs in the recombinant Sf9 cells (Figure 5C) led to a considerable reduction in the contamination of rAAV capsids with *cap*-positive AAV capsids at a level of 0.02–0.03%, but this removal also proved that baculovirus-induced AAV rep/cap template amplification is partially independent of the rep–RBE interaction and, thus, not necessary for rAAV production for this production system. Further optimization of the *cap* sequence construct (splicing-based strategy to raise the relative amounts of VP1 in AAV5 capsids) led to AAV5 infectivity exceeding that of Sf9 or HEK293 cell-based AAV5 production (=development of the OneBac 2.0 system).

The general drawback of the OneBac systems is the fact that, for each new serotype, a new producer cell line has to be generated. A solution was published by Wu et al. [88], who established a packaging cell line which provides only the rep functions, whereas the baculovirus required for the induction of the production of the rAAV vector brings in the *cap* gene of the desired serotype as well as the rAAV genome (ITR-GOI) sequence (=dual functional BEV-CAP-(ITR-GOI)) (Figure 5D). Furthermore, they used a p10 promoter to regulate *cap* gene expression in the baculovirus transactivator, which resulted in lower promoter–promoter interaction (the expression of the rep proteins was under control of the polyhedrin promoter). Although the novel Sf9-GFP/rep packaging cell line-dependent OneBac system is versatile, flexible, and maintains high virus yields, the main drawback is the fact that the *rep* gene construct is inserted into the packaging cell line as performed by Mietzsch et al. [87] for the OneBac 1.0 system. As already mentioned above, the presence of the *RBE* sequence in the construct will still lead to the encapsidation of *rep* sequences into a certain percentage of rAAV particles. With respect to rAAV vector production, this modified OneBac system had the following features: the cells were stable for up to 10 passages, and after infection with rec. baculovirus they showed a specific vector productivity of >10^5^ vg/cell. The serotypes AAV2, AAV8, and AAV9 were produced with this system and the ratio of the capsid proteins had the expected ratio of ~1:1:10 (vp1:vp2:vp3).

Finally, the OneBac system was further optimized by Moreno et al. [89]. The authors kept the separation of the *rep* gene from the *cap* gene by providing the latter gene combined with the rAAV vector construct on one rec. baculovirus (e.g., polH-Cap-ITR-transgene-ITR), more or less similar to that developed by Wu et al. [88]. Concerning the rep gene, a stable Sf9 cell line was constructed containing the inducible *rep* gene (Figure 5E). This article mainly concerned the development of this rep expression cassette. The optimization concerned the choice of baculovirus *hr* sequence, the separation of the large and small *rep* sequences (comparable to Urabe’s approach [45]), and the use of different promoters for driving the expression of the rep proteins. The use of different promoters was chosen in order to avoid promoter interaction related to the expression of the *rep* and *cap* genes and, on the other hand, to obtain to a more sequential expression of the rep proteins. The choice of a double-rep cassette design allowed for an efficient and timely control of rep expression following the expression dynamics of wild-type AAV rep expression. In mammalian cells, the large rep proteins have to be expressed earlier than the small rep proteins because at first rAAV vector DNA has to be replicated (which requires the presence/activity of the large rep proteins). Later on, the small rep proteins are required because they are mainly responsible for the accumulation and packaging of the single-strand vector DNA. Thus, the authors have chosen the 39k promoter, an immediate early promoter, which is transactivated 3–6 h post-baculovirus infection, for driving the expression of the large rep proteins and the polH promoter, a late baculovirus promoter, (or p10 or p6.9) to regulate the majority of expression of the small rep proteins in the double-rep cassette BEV system. This approach led to an increased full-to-empty AAV particle ratio. As mentioned, the authors have also evaluated the use of other baculovirus *hr* enhancer sequences. The original OneBac system used the hr2.09 enhancer, which shows a relatively high basal expression, whereas some alternative enhancer sequences (hr4b, hr5) show reduced basal expression. The authors have shown that all the tested hr sequences enhanced baculovirus promoter activity upon *trans*activation with baculoviruses, and both enhancers (hr4b and hr5) were used in the double-rep cassette in conjunction with the cassette encoding for the large rep proteins and the small rep proteins (in the inverse direction), respectively.

**Figure 5 microorganisms-12-00384-f005:**
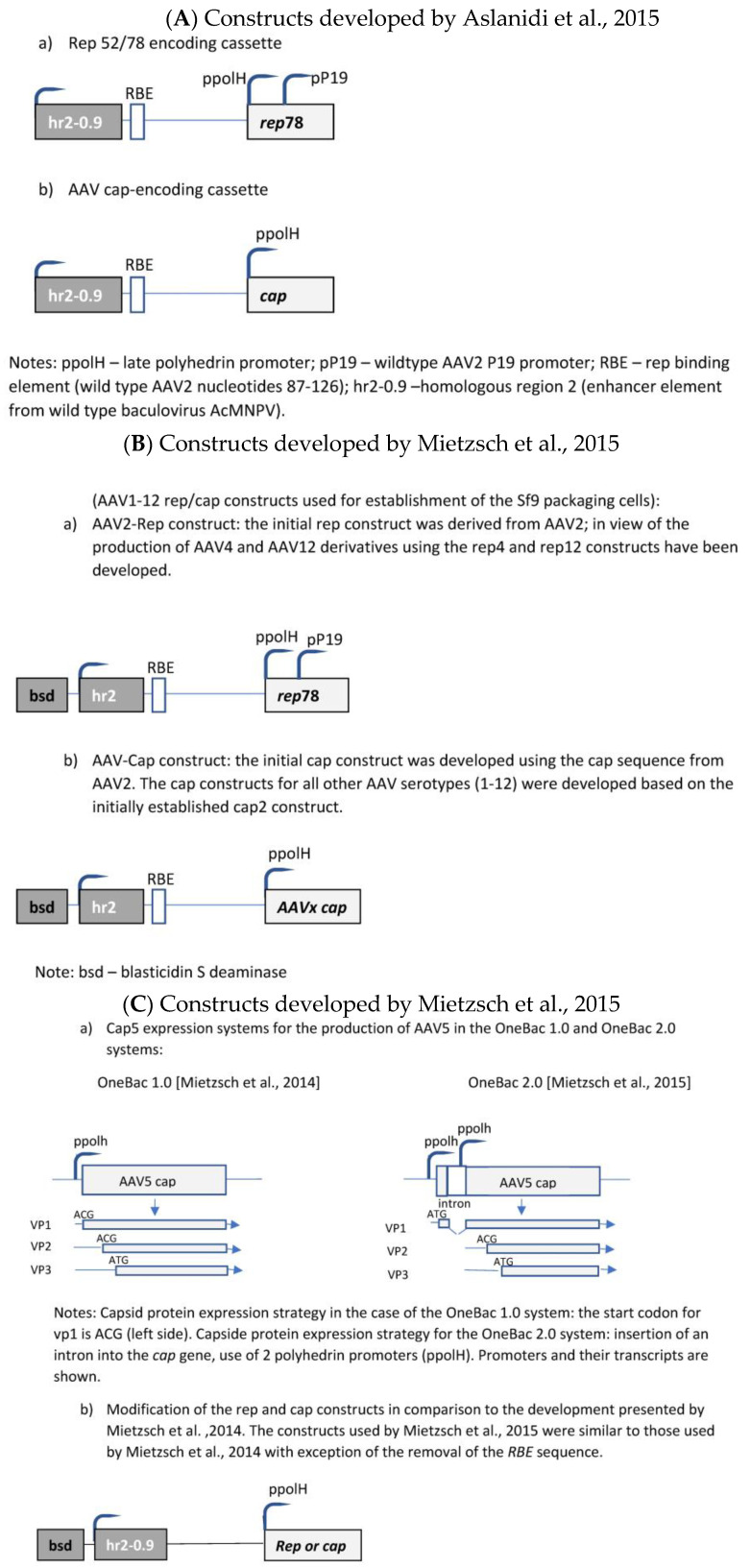
Molecular constructs used for the development of the different insect cell based packaging cell lines. (**A**) Constructs developed by Aslanidi et al. [50]; (**B**) Constructs developed by Mietzsch et al. [87] in the frame of the OneBac development (AAV1-12 rep/cap constructs used for establishment of the Sf9 packaging cells); (**C**) Constructs developed by Wu, Y., et al. [88] in the frame of the OneBac 2.0 development; (**D**) Moreno, F., et al. [89] did not modify the rep construct but developed a combined cap—rAAV construct in order to dispose of a flexible system for easily switching from one AAV serotype to another one and from one transgene construct to another one. Thus, the rep construct was similar to that used by Mietzsch et al. [87]; (**E**) Rep construct developed by Moreno et al. [90].

This system allows for the production of at least 10^11^ gc/mL, with a specific productivity of 10^5^ gc/cell. The final evaluations were performed in 2 L reactors. Furthermore, the authors have shown that the cells were stable for nine passages in the absence of antibiotic selection.

Finally, it should also be mentioned here that because of the way the rep and cap constructs have been developed in frame of the Sf9/baculovirus system, no rcAAV generation is to be expected.

## 4. Comparisons of the Different rAAV Production Systems

Table 1 presents a comparison of the characteristics of different rAAV production systems. Compared are the following features: production scale, specific rAAV production rate, volumetric productivity, rcAAV generation, encapsidation of non-rAAV vector sequences, as well as the percentage of full particles, as far as these data have been published. Many different production systems have been developed or are still under development and optimization, but only some of them are really used for the large-scale production of rAAV vectors: the transfection-based system and the stable producer cell line system have been scaled up to several thousand liters, whereas the ELEVECTA system, HSV-BHK21, and the Sf9 baculovirus systems have been scaled up to at least 100–200 L (as far as has been communicated). The most interesting features, of course, are the specific productivities (10^5^ vg-DRP-gc/c), which directly impact the achievable volumetric titer, the ratio of full-to-empty particle percentage (the higher the better, because the empty particles have to be removed during downstream processing), as well as the absence of rcAAV. Concerning the percentage of full rAAV vector particles, many parameters impact the full-to-empty particle ratio, including rAAV genome length and sequence, ITR integrity, serotype, helper virus, purification method, measurement technique, …), making it difficult to obtain a clear view on the packaging efficiency of each production system. This applies to the HEK293-based production cell lines [76,78,83], the TESSA system (which is also based on the use of HEK293 cells) [65], stable HeLa-based producer cell lines [21,22], and also to improved Sf9/baculovirus systems [49,87,89]. From a safety point of view, the encapsidation of non-rAAV DNA sequences into the AAV particles is also critical and any production system characterized by a reduced encapsidation of such sequences is preferable. Unfortunately, only limited information is available for the different production systems. Concerning the generation of rcAAV, those systems which are characterized by a separation of the rep and cap genes and their lecture in opposing directions (see [90]) do not generate rcAAV. In this context, at least all baculovirus-based production systems as well as all HEK293 cell-based producer cell line systems [65,76,78,83] should be mentioned.

A critical issue which was not discussed in context of this review is the rAAV potency, which is heavily impacted by the biological production system. In this context, Kang et al. [53] have shown that rAAV produced with the HSV1 production system showed a much better in vivo potency for transducing muscles (after intramuscular injection) than rAAV produced by the transfection system. An approximately 5-fold difference was observed. Furthermore, mammalian production systems might be preferable in comparison to an insect cell-based systems because it was shown that the post-transcriptional modifications as well as vector methylation are not identical in mammalian and insect cell systems—rAAV vectors produced by the latter system show a reduced in vivo potency [17].

The ideal system from the author’s point of view is a cell system which allows for the production of at least 10^5^ vg/cell, which does not generate rcAAV, encapsidates few or no foreign DNA sequences into the rAAV capsids, and which is characterized by the use of a chemical induction system. In the case of induction by helper viruses, only one helper virus should be used and, ideally, such a production system should be characterized by a minimal production of helper virus, as shown for the TESSA system [65].

## 5. Conclusions

Over the last 25 years or so, inducible packaging and producer cell lines of rAAV vectors have been developed in order to obtain to a more straightforward production means of these vectors and, thus, reduce overall manufacturing costs. Today, it can be stated that there are three different ‘families’ of such cells: HeLa cell- and HEK293 cell-based producer/packaging cells are routinely used for the large-scale production of rAAV vectors for clinical use. Furthermore, the Sf9 insect cell/baculovirus systems are also used for the large-scale routine production of rAAV vectors for clinical use. In order to be able to produce the required vector amounts, specific productivities of at least 10^5^ vg/cell are required, which are achievable using the current production systems. However, further improvements are required and developments in this direction are ongoing, as actual publications are showing. These developments should/might lead to wild-type AAV-like productivities of about 10^6^ vg/cell, the establishment of intensified production processes using 10–100 times higher production cell densities (as, for instance, shown by Coronel et al. [83] for the ELEVECTA system), as well as to production systems which use only an induction-based approach for rAAV production or a production system which produces only minimal amounts of helper virus as for the TESSA system. These optimization activities should also be extended to the issue of potency, which needs to be increased, in order to reduce the required patient doses. The final aim is obvious: an increase in the rAAV vector productivity will directly lead to a reduction in the production costs and, thus, will make these new treatments available to a larger patient population.

## Figures and Tables

**Figure 1 microorganisms-12-00384-f001:**
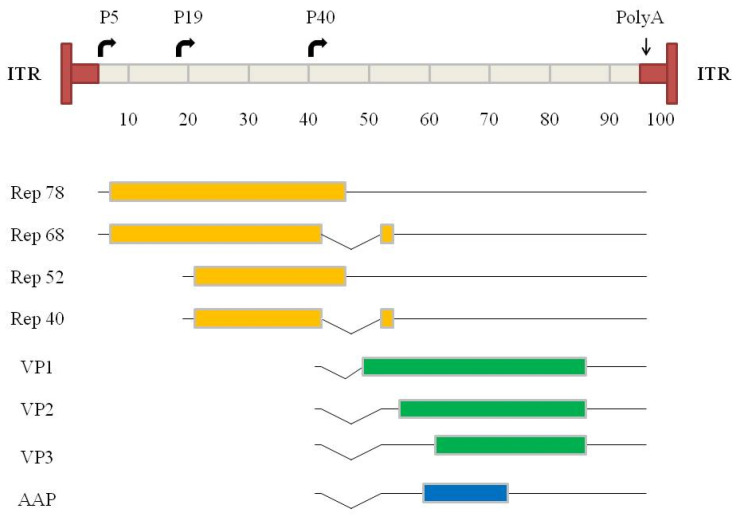
Genome organization of wild-type AAV. A scale of 100 map units is used (equivalent to about 4700 nucleotides). (**Top**) T-shaped dark boxes indicate the ITRs. The horizontal arrows indicate the three transcriptional promoters (P5, P19, P40). (**Bottom**) The solid lines indicate the transcripts, and the introns are shown by the broken lines. A poly-adenylation signal present at map position 96 is common to all transcripts. The first ORF encodes the four regulatory proteins (rep 78, rep 68, rep 52, rep 40) for which transcripts arise from the promoters P5 and P19 in combination with alternative splicing. The second ORF driven by promoter P40 encodes the three capsid proteins (VP1, VP2, VP3) from two transcripts. VP1 is initiated from the first cap transcript, and VP2 and VP3 are translated from two different start codon sites from the second cap transcript. Note that the translation initiation site of VP2 is an ACG. The second cap transcript encodes the assembly-activating protein (AAP), necessary for AAV capsid assembly (from Galibert and Merten [24]).

**Figure 2 microorganisms-12-00384-f002:**
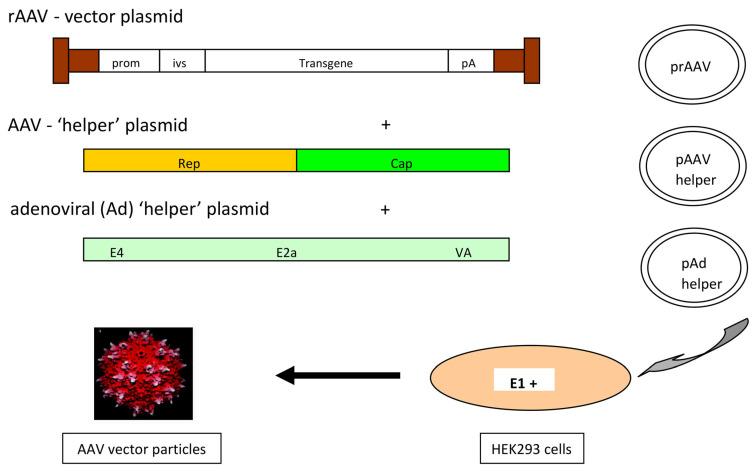
rAAV design and production principle (in a triple-transfection setting). AAV helper (rep, cap), rAAV vector, and the adenoviral helper (E2A, E4orf6, VA RNA genes) plasmids are brought into HEK293 cells (which are E1A, E1B positive) by Ca-phosphat or PEI transfection. Notes: prom—promoter; ivs—intervening sequence (e.g., intron); pA—polyA sequence.

**Figure 3 microorganisms-12-00384-f003:**
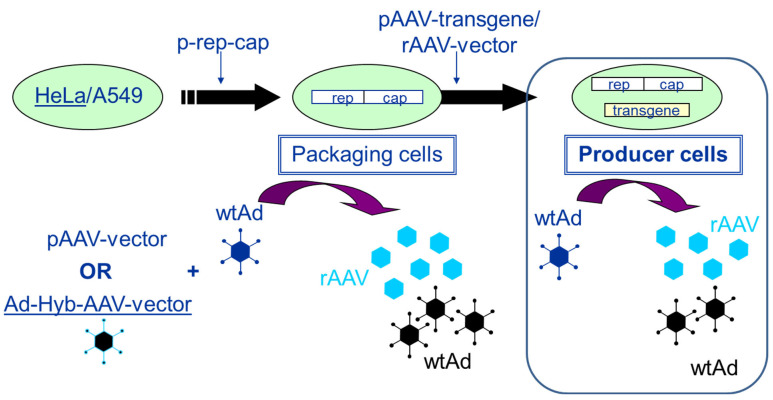
Establishment of packaging and producer cell lines for rAAV vector production. The transfection of HeLa or A549 cells with a plasmid harboring the rep2 (from AAV2)-capx (from any AAV serotype) sequences (p-rep-cap in the figure) leads to the establishment of a packaging cell line. When these cells are either transfected with a pAAV-vector (plasmid) followed by infection with an helper virus (wtAd) orinfected with adenovirus (wtAd) and 24 h later with an E1-deleted adenovirus—AAV-Hybrid virus (providing the rAAV transgene sequence flanked by the two ITRs) (Ad-Hyb-AAV-vector)—rAAV as well as adenovirus are produced. If the rAAV transgene sequence is stably integrated via plasmid transfection (pAAV-transgene) or vector transduction (rAAV-vector), the packaging cells become stable producer cells. Upon infection with adenovirus (wtAd), these cells start to produce rAAV vector and adenovirus (wtAd).

**Table 1 microorganisms-12-00384-t001:** Comparison of different rAAV vector production systems.

	HEK293 Transfection	HEK293 Inducible Producer Cell Line	HEK293 Inducible Producer Cell Line	ELEVECTA (HEK293/CAP)	HeLa—Packaging Cell Line	HeLa—Producer Cell Line	TESSA System	rHSV Based Infection System	Baculovirus System (Sf9, 2 Viruses)	Monobac System	OneBac 2.0 System	Optimized Onebac System
Cell line	Selected clone of wildtype HEK293	HEK293 containing all required functions, activation of rep functions after AdV infection	HEK293 containing all required functions	HEK293 or CAP cells containing all requried functions	HeLa containing *rep* & *cap* genes	HeLa containing *rep* & *cap* genes and rAAV vector construct	Wildtype HEK293	sBHK21	Wildtype Sf9 cells	Wildtype Sf9 cells	Sf9 cells containing the *rep* & *cap* genes, inducible upon baculovirus infection	Sf9 cells containing the *rep* genes, inducible upon baculovirus infection
Induction of rAAV production	Transfec-tion using plasmids	Infection with AdV carrying the *cre* gene for inducing *rep* gene expression	Dual induction of gene expression using doxycycline & cumate	Induction using doxycycline	Infection with wtAdV & rAAV hybrid AdV	Infection with wtAdV or attenuated AdV or with HSV	Infection with 2 TESSAs delivering *rep/cap* genes & rAAV vector construct	Infection with 2 HSV1 viruses delivering *rep/cap* genes & rAAV vector construct	Infection with 2 baculovirus delivering *rep/cap* genes & rAAV vector construct	Infection with 1 baculovirus delivering all required functions	Infection with 1 baculovirus delivering the rAAV vector construct	Infection with 1 baculovirus delivering the *cap* genes and the rAAV vector construct
Large scale	Several 1000 L ^(1)^	Research grade	Research grade	At least 200L	Small scale	2000 L	Large scale	10–100L (WAVE)	200L (STR, WAVE)	Small scale (2L STR)	Small scale	Probably developed for large scale production
Cell specific pro-ductivity	1–2 × 10^5^ vg/c (AAV2)	0.9–1.3 × 10^5^ vg/c	2.5 × 10^3^−1.6 × 10^4^ vg/c	No infor-mation	2.2 × 10^4^–5.9 × 10^5^ gc/c	>5 × 10^4^–>10^5^ DRP/c	No infor-mation	5.5 × 10^4^–1.3 × 10^5^ DRP/c (AAV8)	3.7–9.6 × 10^4^ vg/c	Up to 5 × 10^5^ vg/c (AAV2)	1–2 × 10^5^ vg/c (AAV5)	~10^5^ gc/c
Volumetric production	10^14^ vg/L	9 × 10^13^–1.3 × 10^14^ vg/L ^(6)^	10^13^ vg/L	10^14^ vg/L–2 × 10^16^ vg/L ^(7)^	2.2 × 10^13^–5.9 × 10^14^ gc/L	>5 × 10^13^ vg/L	2 × 10^13^ gc/L	2.4 × 10^14^ DRP/L	3–4 × 10^15^ vg/L	6–7 × 10^15^ vg/L	1.4 × 10^15^ vg/L	~10^14^ gc/L
rcAAV production	+ ^(2)^	No rcAAV/10^9^ rAAV-vg	Probably no because *rep* and *cap* genes are separated with opposite transcriptional orientations	No infor-mation	No rcAAV/10^9^ rAAV-vg	<0.0002%	No infor-mation	No (below detection limit)	No (below detection limit)	No (below detection limit)	No (below detection limit)	No infor-mation
Full/empty particle ratio	5–50%	No infor-mation	24–56% ^(5)^	28–35%	No infor-mation	>50%–>70%	15–20%	5.5–8.3%	10–40% ^(8)^	36%	No infor-mation	No infor-mation
Encapsida-tion of helper sequences	Rep: 0.3–1.5% ^(3)^, Cap: 0.4–1% ^(4)^	Not communi-cated	Not communi-cated	Not communi-cated	Not communi-cated	0.02–0.05% (rep, cap)	~2.5% rAAV contain AdV packaging signal ^(9)^	HSV: 0.007–0.012%	Cap: 0.016%, Rep: 0.019–0.014%	Not communi-cated	Cap: 0.02%, Rep: <0.001%	Should be better than or equal to the OneBac 2.0 system
Reference	Grieger et al. [23]	Yuan et al. [75]	Lu et al. [78]	Coronel et al. [83]	Gao et al. [62], Zhang et al. [63]	Thorne et al. [21], Martin et al. [22]	Su et al. [65]	Thomas et al. [20], Clément et al. [91], Kang et al. [53]	Smith et al. [48], Dickx et al. [92], Galibert et al. [49]	Galibert et al. [49], Merten [93]	Mietzsch et al. [87]	Moreno et al. [89]

Notes: (1) Grieger et al. [23] published results for cultures done in a 20L WAVE reactor. (2) rcAAV generation depends on the plasmid system used (compare with Emmerling et al. [90]). (3) Chadeuf et al. [94]. (4) Gao et al. [95]. (5) full to empty particle ratio depended on the clones analyzed. (6) calculated via a theoretical cell density of 10^6^ c/mL. (7) Perfusion culture at 2 × 10^7^ c/mL. (8) depending on the ITR construct [96]. (9) in the second production cycle using rAAV for infecting HEK293 cells, this percentage is reduced to 0.05% [65]. Abbreviations: vg—vector genome, DRP—DNAse resistant particle, gc—genome copy.

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
