# Peer review of "Development of Stable Packaging and Producer Cell Lines for the Production of AAV Vectors"

_microorganisms, 2024, doi:10.3390/microorganisms12020384_

Round 1

Reviewer 1 Report

Comments and Suggestions for Authors

In this Review, Dr Otto W Merten has extensively analyzed the technologies available to generate packaging and producer cell lines for manufacturing AAVs. The review is very comprehensive and accurate, particularly detailed on the molecular biology aspects. 

I have only few suggestions to make the article more reader friendly and give more context to some of the statements, in particular:

-In the introduction, it could be nice to make some forecast of AAV vector needs in the future and why and how packaging/producer cell lines might become the preferred options or not compare to other alternatives

-I propose to split Fig2 A, B in two figures to match easily the text (make Fig 2 and Fig 3 respectively from A and B)

-Include a figure for Sf9 cell lines will be also helpful for the reader to better visualize all genetic elements involved in this system

-Together with the conclusions, a summary table including the technology (HEK, Hela, Sf9, etc), vector yields, scale, infectivity, contaminants, and/or other attributes would be extremely useful. It is understood that not all information will be available for all systems, but still will be very informative.

-In conclusions sections, a more conscious comparison between producer/packaging cell lines with other transient systems (transfection, HSV, TESSA,...) would be appreciated to put more context/value for this review

Author Response

Dear Reviewer,

Thank you very much for taking the time to review this manuscript. Please find the detailed responses in the attached file  and the corresponding revisions/corrections highlighted in the re-submitted files.

Reviewer 2 Report

Comments and Suggestions for Authors

In this review, the authors primarily discuss the progress in developing stable adeno-associated virus (AAV) packaging and production cell lines. They outline the characteristics of packaging and production cell lines based on HeLa cells, HEK293T cells, and Sf9 cells. However, there are several areas that require improvement:

1. The main focus and key points of the review are not clearly emphasized throughout the article. The titles of each section should be distinguished as either main or secondary, in order to establish a clear and logical structure. Additionally, the overall readability of the paper needs improvement as it is currently confusing.

2. The author focuses on "in vivo treatment based on AAV vectors" at the beginning of the introduction, and discusses it in large lengths, lacking background knowledge on adeno-associated viruses.

3. As a review paper, the number of references should be increased to demonstrate a better understanding of the field and enhance the quality of the review. Furthermore, the references cited in the article are relatively dated and should be updated to include more recent and relevant research.

4. The diagrams in the article are highly irregular, particularly Figure 2. The text overlaps and is misaligned, while the connections between straight lines and arrows are not smooth. It is important to create standardized and visually clear diagrams.

5. Figure3A, figure3B, and figure3C mentioned in the text do not show the relevant figures. The pictures and texts are inconsistent, please verify.

6. The abbreviations used in the article are inconsistently and irregularly formatted. Many abbreviations are used without providing their full English names. It is essential to consistently adhere to a standard format for abbreviations throughout the article.

7. The formatting of gene names in various sections of the article is inconsistent. Some gene names are italicized, while others are not. It is important to apply a consistent formatting style for all gene names.

8. The language throughout the entire paper requires further editing and polishing to improve the overall clarity and readability.

Author Response

(The authors gave the same response as above.)

Reviewer 3 Report

Comments and Suggestions for Authors It is appreciate that the author tried to summarize the development of different AAV producer cell lines, however this review article is hard to follow, and the content appears scattered. As a review, it lacks author’s view and comments to help the readers to use the information provided, especially for novices in the field. I weight this a bit more for a good review.

The content of this review article lacks a comprehensive comment on the developments, especially the advantage and disadvantage in using the rAAV producer cell lines.  

Author Response

(The authors gave the same response as above.)

Round 2

Reviewer 2 Report

Comments and Suggestions for Authors

-

Author Response

The reviewer has no further suggestions